# Empirical Evaluation of Pretraining Strategies for Supervised Entity Linking

**Thibault Févry**[*†]                    TFEVRY@GOOGLE.COM

**Nicholas FitzGerald***                    NFITZ@GOOGLE.COM

**Livio Baldini Soares**                    LIVIOBS@GOOGLE.COM

**Tom Kwiatkowski**                    TOMKWIAT@GOOGLE.COM

*Google Research, NYC*

## Abstract

In this work, we present an entity linking model which combines a Transformer architecture with large scale pretraining from Wikipedia links. Our model achieves the state-of-the-art on two commonly used entity linking datasets: 96.7% on CoNLL and 94.9% on TAC-KBP. We present detailed analyses to understand what design choices are important for entity linking, including choices of negative entity candidates, Transformer architecture, and input perturbations. Lastly, we present promising results on more challenging settings such as end-to-end entity linking and entity linking without in-domain training data.

## 1. Introduction

Traditionally, entity linking approaches have relied on knowledge bases, complicated modelling and task-specific hand-engineered features to achieve high performance. More recently, Broscheit 2019, Ling et al. 2020 and Wu et al. 2019 show that using large-scale pretrained language models like BERT [Devlin et al., 2018], pretraining on Wikipedia entity links, and fine-tuning on a specific entity linking corpus leads to state-of-the-art performance without relying on such features. However, Ling et al. 2020 focused mainly on constructing general-purpose entity representations, Wu et al. 2019 on building strong zero-shot entity linking systems, and Broscheit 2019 on end-to-end linking, so that the limits of pretraining for entity disambiguation have not been fully explored.

In this paper we present a thorough study of pretraining strategies for supervised entity linking. We establish new upper bounds for performance on the widely studied CoNLL and TAC-KBP 2010 entity linking tasks. We also show that our pretraining approach yields a very competitive entity linking system without any further domain specific tuning. We present a detailed analysis of significant design choices including the choice of negative candidates used during training, and the document context encoded for each mention. We find that the optimal choice of negative candidates is dependent on whether or not the final linking system has access to an alias table. For a system that will use an alias table at inference, it is helpful to pretrain the models with lexically similar candidates. However, when no alias table is used for the downstream task, ensuring candidates are random improves the model ability to distinguish the right entity among all possible entities.

---

*. Denotes equal contribution.

†. Work conducted during Google AI Residency.

In our studies we found two surprising results: (1) it is possible to achieve optimal entity linking results with a four layer transformer, which is one third of the size of BERT-base, and (2) given the abundance of supervision from Wikipedia links, we did not get any gains in performance from training with an auxiliary language modeling loss. However, we did find that the input perturbations introduced by [Devlin et al., 2018] themselves increase the quality of our pretraining approach and we present an analysis of how these perturbations add robustness to the model.

Finally, to demonstrate the generality of our model, we present results on the end-to-end entity linking task in which both mention location and identity are predicted. On this task, our model outperforms all but the tailored methods introduced by Kolitsas et al. 2018, Broscheit 2019. We argue that our model is more practical, and easier to integrate, than heavily engineered existing approaches, and we believe that downstream tasks such as information extraction and question answering can benefit from this robust standalone end-to-end entity linker (e.g. [Févry et al., 2020]).

## 2. Related Work

Early entity linking systems [Bunescu and Pasca, 2006, Mihalcea and Csomai, 2007] focused on matching the context of the mention with that of the entity page. In addition to context features, systems have relied on $\mathbb{P}(e|m)$, the prior probability that mention $m$ refers to entity $e$, computed from Wikipedia mention counts. The set of entities in a document should be globally coherent, and several approaches have introduced sophisticated global disambiguation methods [Globerson et al., 2016, Cheng and Roth, 2013, Sil and Yates, 2013] that consider all mentions in a document to make predictions. In contrast, we do not model document-level disambiguation explicitly. However, our long context windows contain several mentions which should allow such disambiguation. Other approaches have also sought to incorporate types and knowledge base information in their modelling, such as Radhakrishnan et al. 2018, Raiman and Raiman 2018.

We pretrain our model by learning distributed representation of entities directly from Wikipedia text, similarly to Yamada et al. 2016, 2017, Gillick et al. 2019, Ling et al. 2020. Unlike Wu et al. 2019, Gupta et al. 2017, our embeddings are learned directly rather than generated through entity descriptions. In contrast with Yamada et al. 2016, 2017, we do not use additional features, such as prior probabilities or string match features. Our method is therefore most similar to Ling et al. 2020 and Broscheit 2019 who also use a transformer. We differ from Ling et al. 2020 by simultaneously considering all mentions and entities in a context and, in contrast to both, only use a four layer, randomly-initialized transformer instead of twelve layers initialized from large scale language modelling pretraining.

In addition, we experiment with end-to-end entity linking [Sil and Yates, 2013, Luo et al., 2015, Kolitsas et al., 2018], where instead of predicting the entity for gold spans, a system must both predict the span and its label. A closely related task is multilingual entity linking. Approaches have used multilingual embeddings to link text in several languages [Sil et al., 2018, Tsai and Roth, 2016]. Zero-shot entity linking [Logeswaran et al., 2019, Wu et al., 2019] is another relevant task. In that setting, entities predicted at test time are not seen in training. Instead, the system relies on the entity name and description.

## 3. Model

### 3.1 Task Definition

Let $\mathcal{E} = \{e_1 \dots e_N\}$ be a predefined set of entities, and let $\mathcal{V} = \{[\text{MASK}], w_1 \dots w_M\}$ be a vocabulary of words. A *context* $\mathbf{x} = [x_0 \dots x_t]$ is a sequence of words $x_i \in \mathcal{V}$. A *span* $\mathbf{s} = (s_{start}, s_{end})$, is a tuple with $0 \leq s_{start}, s_{end} < t$ which defines a contiguous sequence of tokens in a given context. A *mention label* $\mathbf{l} = (s_k, e_k)$ consists of a span $s_i$ and an entity label, $e_i \in \mathcal{E} \cup \varnothing$. We use $\bar{\mathbf{l}}$ to denote a set of such mention labels. The NULL-symbol $\varnothing$ indicates a span that is labeled as a mention, but without an entity linking label.

Our training data, $\mathcal{D} = \{(\mathbf{x}_0, \bar{\mathbf{l}}_0) \dots (\mathbf{x}_N, \bar{\mathbf{l}}_N)\}$, is a corpus of contexts, each paired with a *set* of mention labels, one for each mention in the context. Given an input context $\mathbf{x}_i$, our goal is to predict the set of entity mentions $\bar{\mathbf{l}}_i$. In *Entity Disambiguation*, we are given the set of spans, and predict the entity linked by each span. In *End-to-End Entity Linking*, we must predict both the set of mention spans, and their linked entities.

### 3.2 Contextual Language Representation

Our model is built using the now-standard Transformer-based architecture [Vaswani et al., 2017]. The model computes a matrix representation $\hat{\mathbf{H}} \in \mathcal{R}^{t \times d}$ of a text sequence through successive application of a Transformer block to the output of the previous layer:

$$\mathbf{H}_i = \texttt{TransformerBlock}(\mathbf{H}_{i-1})$$
$$= \texttt{MLP}(\texttt{MultiHeadAttention}(\mathbf{H}_{i-1}, \mathbf{H}_{i-1}, \mathbf{H}_{i-1}))$$

$\mathbf{H}_0$ is a sequence of context-independent token embeddings and $\hat{\mathbf{H}} = \mathbf{H}_n$, where $n$ is the number of Transformer layers.

### 3.3 Entity Disambiguation

Each entity $e \in \mathcal{E}$ is mapped directly onto a dedicated vector in $\mathbb{R}^d$ via a $|\mathcal{E}| \times d$ dimensional embedding matrix. In our experiments, we have a distinct embedding for every concept that has an English Wikipedia page, resulting in approximately 5.7m entity embeddings.

In order to perform entity linking for a particular span with word-piece token indices $(i, j)$, we (following Lee et al. 2016) first obtain a representation of the span by concatenating the representation at the span start and end, and pass this through a multi-layer perceptron which projects the span representation into the same space as the entity embeddings.

$$\hat{s}_{\mathbf{s}_i} = \texttt{MLP}([\mathbf{H}_{n,s_{start}}, \mathbf{H}_{n,s_{end}}]) \tag{1}$$

Our model scores each span-entity pair by taking the dot-product between the projected span representation and the embedding of $\mathbf{e_c}$. Thus, the conditional probability that the span $\mathbf{s}_i$ refers to entity $e_c$ is defined as:

$$\mathbb{P}(e_c | \mathbf{s}_i) = \frac{\exp(\hat{s}_{\mathbf{s}_i} \cdot e_c)}{\sum_{\mathbf{c}' \in \mathcal{E}} \exp(\hat{s}_{\mathbf{s}_i} \cdot e_{c'})} \tag{2}$$

In practice, this is expensive to compute for large $|\mathcal{E}|$. Therefore, for every $\bar{\mathbf{l}}$ we select a set $\mathcal{C}_{\bar{\mathbf{l}}}$ of $k$ candidates, which contains the entity labels for all $\mathbf{l} \in \bar{\mathbf{l}}$ as well as a set of negative candidates. We do not have an entity linking loss on mentions that do not have a label. Therefore, our per-example entity linking loss is:

$$l_{linking}(\bar{\mathbf{l}}) = \sum_{\mathbf{l}_i \in \bar{\mathbf{l}}} \frac{\exp(\hat{s}_{\mathbf{s}_i} \cdot e_i) \mathbb{1}_{e_i! = \varnothing}}{\sum_{c \in \mathcal{C}_{\bar{\mathbf{l}}}} \exp(\hat{s}_{\mathbf{s}_i} \cdot e_c)} \tag{3}$$

We will discuss further how we select $\mathcal{C}_{\bar{\mathbf{l}}}$ in Section 4.1.

### 3.4 Mention Detection

For many entity linking tasks, the target spans are provided. In order to be able to do end-to-end entity linking, we additionally train our model to predict mentions, independently of entity linking. One way to do this would be to score every possible span-entity pair, and either use a score threshold to filter spans where no entity link achieves a sufficiently high score, or to additionally score a special NULL-link embedding. However, enumerating all spans for the long contexts we use in our model would be prohibitively expensive. We take the approach of encoding mentions as a BIO sequence, and train an MLP on the context representation to predict this sequence with a standard cross-entropy loss. Our final loss sums the mention detection loss and the linking loss.

## 4. Experimental Setup

### 4.1 Wikipedia Pretraining

We build a training corpus of contexts paired with entity mention labels from the 2019-04-14 dump of English Wikipedia. We first divide each article into chunks of 1000 unicode characters, resulting in a corpus of over 17.5 million contexts with over 17 million entity mentions covering over 5.7 million entities. These are processed with the BERT tokenizer, limited to 256 word-piece tokens. In addition to the Wikipedia links, we annotate each sentence with unlinked mention spans using a state-of-the-art named entity recognizer. These are used as additional signal for our mention detection component.

**Entity Candidates Selection**    Training the model with a full softmax over all 5.7 million entities for every mention is computationally expensive. A common solution is to use a noise contrastive loss [Gutmann and Hyvärinen, 2012, Mnih and Kavukcuoglu, 2013] and sample candidates according to their relative frequency, as in Ling et al. [2020]. In this work, we experiment with other approaches to candidate generation that might provide better negatives in training. In addition to negatives selected uniformly at random from the entire entity vocabulary, we define two types of hard negatives:

1. **Page candidates**, which is the set of all entities linked to in the article from which the given context was taken. This is meant to capture semantically related concepts.

2. **Phrase table candidates**, the set of lexically related entities for each mention candidate, obtained from the Phrase Table provided by SLING [Ringgaard et al., 2017].

Throughout the paper, we will use $|\mathcal{C}_{\bar{1}}| = 768$. In our base setup, we use up to 256 page candidates, and 384 phrase table candidates, equally divided between each mention in the example. Any remaining room in the set of 768 is filled with random candidates sampled uniformly from the entity vocabulary (meaning a minimum of 128 random candidates per example). We will study the impact of different candidate selection methods in Section 6.2. In addition to those candidates, for every example in a batch, we use the candidates of other examples as additional negatives.

**Input Noising**  We also add noise to the input data. We apply the same noise function as is used in Devlin et al. 2018: 15% of the tokens are chosen to be modified. 80% of those tokens are changed to the [MASK] token, 10% are changed to a random token and 10% are left unmodified.

**Pretraining hyperparameters**  We use ADAM [Kingma and Ba, 2014] with a learning rate of 1e-4 to optimize our model. We use a linear warmup schedule for the first 10% of training, decay the learning rate afterwards and use gradient clipping with a norm of 1.0. We train from scratch for up to a million steps and use a large batch size of 8192 for pretraining. We follow BERT [Devlin et al., 2018] base for many of our model parameters, though we do not use large-scale language-modeling pretraining and only use four layers, as we did not find more to layers to further improve performance. We use the same word-piece vocabulary as the lowercase version of BERT. We use entity embeddings of size 256 unless mentioned otherwise. We weight both the entity disambiguation loss and the mention detection loss to 1. We use a context window of 256 tokens.

## 4.2  Entity Linking Datasets

We evaluate our model on two popular entity linking benchmarks: AIDA CoNLL YAGO [Hoffart et al., 2011] and TAC-KBP 2010 [Ji et al., 2010]. The first is comprehensively annotated with approximately 34,000 mentions on 1,393 newswire document on the full Wikipedia vocabulary, while the second is sparsely annotated for target entities only on a smaller entity vocabulary.

### 4.2.1  AIDA CoNLL-YAGO DATASET

**Textual Context**  Most CoNLL documents do not fit in our limit of 256 tokens. Therefore, we split the document into "sentences" at each newline in the document. We experiment with three methods to add document context to these sentences: (i) taking the sentence as-is, (ii) adding the title of the document to the sentence, (iii) adding the title of the document as well as the first two sentences to the sentence. Throughout our experiments we will use (iii), though we show the impact of this choice in Section 6.4.

**Entity Candidates Selection**  Our candidates for CoNLL come from alias tables - resources which provide a list of possible strings for a given entity. A key challenge with evaluating entity linking systems on the CoNLL dataset is inconsistent use of alias tables. Globerson et al. 2016 describe the difficulty of resolving older resources due to changes in Wikipedia links and unicode, and provide statistics for two commonly-used alias tables: The YAGO extended "means" mapping of  Hoffart et al. 2011, and the "PPRforNED" mapping of  Pershina et al. 2015.

| Alias Table | Conversion | Gold recall | Avg. ambig. |
|---|---|---|---|
| Hoffart et al. 2011 | Globerson et al. 2016 | 96.19 | 65.9 |
| | Ours | 99.33 | 67.4 |
| Pershina et al. 2015 | Globerson et al. 2016 | 99.84 | 12.6 |
| | Ours | 99.75 | 13.8 |

Figure 1: Statistics for alias table conversions, computed on the CoNLL test split. Gold recall is the percentage of mentions for which the gold entity is included in the candidate set. Average ambiguity is the total number of candidates divided by the number of mentions.

We find that through careful resolution of unicode and Wikipedia redirects, we achieve a slightly higher conversion rate than reported by Globerson et al. 2016 (statistics provided in Table 1). This leads to a higher gold recall, but also a larger number of candidate for each mention, meaning our system must distinguish between more candidates. We report results using both alias tables.

**Finetuning** We finetune our model – including the entity embeddings – on the CoNLL training set, using the alias table candidates for each mention. We used a batch size of 256, a learning rate of 1e-6, and train for 2000 steps.

### 4.2.2 TAC-KBP 2010 DATASET

TAC-KBP 2010 is another widely used dataset for evaluating entity disambiguation systems. In contrast with CoNLL, the mentions are sparsely annotated among documents. It contains 1074 annotated entities in the training set and 1020 in the evaluation set. The entities for this dataset are part of the TAC Knowledge Base, containing 818,741 entities. Due to the reduced entity vocabulary, we can fine-tune without resorting to an alias table and we adopt this setting throughout our results. This is consistent with the prior state-of-the-art approach of Wu et al. 2019. To select the context for a mention, we take the 256 bytes before and after the first occurrence of the mention in the document.

We select the fine-tuning parameters on training by doing cross-validation on the training set. We used a batch size of 32, trained for 1,000 steps and found it was best to freeze the entity embeddings. Our final model is trained on all the training data with the parameters selected in cross-validation. However, we report the result on the evaluation (test) set number in all tables, including ablations. Indeed, we found that this was more reflective of task performance, as the training set is significantly easier.

### 4.3 End-to-end entity linking

We also experiment with end-to-end entity linking on CoNLL (TAC-KBP is not suitable due to its sparse annotations). In this case, we do not use an alias table. In this setting, we follow the hyperparameters of Section 6.1. Instead of using candidates, we train our model to predict BIO-tagged mention boundaries and to disambiguate among all entities. At training and fine-tuning time, gold spans are used for the disambiguation task. At

| System | CoNLL H | CoNLL P | TAC-KBP 2010 |
|---|---|---|---|
| Chisholm and Hachey 2015 | 88.7 | - | 80.7 |
| Ganea et al. 2016 | 87.6 | - | - |
| Globerson et al. 2016 | 91.0 | - | 87.2 |
| Pershina et al. 2015 | - | 91.8 | - |
| Globerson et al. 2016 | - | 92.7 | 87.2 |
| Yamada et al. 2016 | 91.5 | 93.1 | 85.2 |
| Raiman and Raiman 2018* | | 94.9 | 90.9 |
| Yamada et al. 2017 | - | 94.3 | 87.7 |
| Ling et al. 2020† | - | 94.9 | 89.8 |
| Wu et al. 2019 | - | - | 94.0 |
| OURS† | **92.5** | **96.7** | **94.9** |

Table 1: Test accuracy on the CoNLL and TAC-KBP entity disambiguation tasks. CoNLL H refers to papers using the Hoffart et al. 2011 "means" alias table while P refers to the Pershina et al. 2015 table. *It is not clear which alias table, if any, is used by Raiman and Raiman 2018. †-marked systems do not use features beyond the text and alias table.

inference, we use the BIO-tagged predictions as our spans and predict entities for each span among all possible entities. We use the standard strong matching micro-F1 score.

## 5. Evaluation

### 5.1 Entity Linking

Table 1 shows that our approach outperforms all prior approaches on CoNLL and TAC-KBP 2010. On CoNLL, we outperform methods in both alias-table settings. Additionally, we note that unlike many previous systems, we do not use alias priors, knowledge-base features, or other entity features.

### 5.2 End-to-end entity linking

For end-to-end entity linking, we do not use the alias table. Instead of using candidates, we predict BIO-tagged mention boundaries and disambiguate mentions among all entities. At training and fine-tuning time, gold spans are used for the disambiguation task. At inference, we use the BIO predictions as our spans and predict entities for all these spans.

Table 2 shows our model fares well against other models, with the exception of Broscheit 2019 and Kolitsas et al. 2018. The former use a much larger Transformer model, and also initialize from BERT-base model, which is pretrained on a corpus of unlabeled text much larger than our training data. Kolitsas et al. 2018 relies on an alias table to generate candidate mentions at both training and inference time. In addition, it introduces a clever mechanism to jointly optimize and select mention boundaries and entity candidates, whereas we use a simpler pipelined approach. Finally, they also introduce a document-level disambiguation coherence penalty and a coreference resolution heuristic. We believe the use of an alias

| System | Development | Test |
|---|---|---|
| Daiber et al. 2013 | 55.2 | 57.8 |
| Hoffart et al. 2011 | 72.4 | 72.8 |
| Piccinno and Ferragina 2014 | 72.8 | 73.0 |
| Peters et al. 2019 | 82.1 | 73.7 |
| Broscheit 2019 | 86.0 | 79.3 |
| Kolitsas et al. 2018 | 86.6 | 82.4 |
| OURS | 79.7 | 76.7 |

Table 2: End-to-end entity linking strong matching micro-F1 score on the development and test sets of CoNLL. Despite the simplicity of our setup, our system is competitive with most prior work, with the noteworthy exception of Kolitsas et al. 2018

.

| Setup | No fine-tuning | Fine-tuned |
|---|---|---|
| Pre-trained and fine-tuned with all entities | 85.9 | 91.7 |
| No pre-training, fine-tuned with candidates | - | 88.4 |
| Pre-trained with all entities, fine-tuned with candidates | 92.4 | 97.2 |
| Pre-trained and fine-tuned with candidates | 92.6 | 97.1 |

Table 3: Entity disambiguation accuracy on the CoNLL development set for different pre-training and fine-tuning setups. Numbers in the first part of the table do not use an alias table, whereas the ones on the second part use Pershina et al. 2015's table.

table as well as the aforementioned differences explain the gap between our method and Kolitsas et al. 2018, and we will look to bridge this gap in future work. Nevertheless, our model stands as a strong baseline of what can be achieved with simple modelling and low inference cost.

## 6. Analysis

### 6.1 Classifying all entities or classifying candidates

We trained our model to distinguish the correct linked entity among candidates. An alternative approach is to predict among all entities. This is computationally more expensive as it requires doing a softmax over 5.7 million entities for every mention in the batch. Thus we use a batch size to 2048 and set the entity embedding dimension to 64 for both this model and the one trained with candidates. Table 3 shows impressive accuracy without an alias table for the system classifying among all entities. However, it does not fare better than the model trained with candidates when using one. Given the considerable cost of doing the full sofmax for every mention, we use candidates for our other experiments. Note that our model gets 88.4% accuracy when trained only on the CoNLL data.

|  | CoNLL | | TAC-KBP | |
| --- | --- | --- | --- | --- |
| Candidates source | No fine-tuning | Fine-tuned | No fine-tuning | Fine-tuned |
| OURS | 92.2 | 96.9 | 87.3 | 91.4 |
| Phrase table and random | 88.0 | 96.9 | 91.6 | 94.7 |
| Page and random | 83.4 | 96.9 | 92.4 | 94.4 |
| Random | 85.3 | 97.2 | 91.7 | 94.9 |

Table 4: Impact of the candidate selection method on development performance on CoNLL and TAC-KBP 2010. Unsurprisingly, pretraining methods that are closer to the final task setup perform better: For CoNLL, we emphasize the importance of phrase table candidates in pre-training to emulate the use of the alias table , whereas for TAC-KBP, setups that use random candidates are more successful as they are closer to the full classification setup used in this task.

## 6.2 Impact of candidate selection

In Table 4, we show the impact of different candidate selection pretraining strategies. On CoNLL, where we do use an alias table for evaluation, we find that our candidate selection heuristic seems to help the model in pre-training, achieving better performance than any of our ablations. Training with lexically related (phrase table) candidates is particularly important, as this is similar to the disambiguation task the model has to perform. Page candidates are semantically related but generally not lexically related and thus do not bring the same benefits. In fact, they might even distort the distribution of negative candidates as the model performs worse in this setting than with random candidates. After fine-tuning, all models fare similarly, which we believe is due to CoNLL having enough fine-tuning data so that all our models approach the performance upper-bound on this task (see Section 6.5).

For TAC-KBP 2010, where we do not use an alias table at fine-tuning or inference time, the results are markedly different. After pre-training, OURS which has less random candidates performs worse than all other alternatives. This is likely because having more random candidates is closer to the full classification setup used in TAC-KBP 2010. Given TAC-KBP's small training set, these differences carry over in fine-tuning performance.

## 6.3 Impact of adding noise during pretraining

Table 5 shows adding noise in pretraining helps both pre-training and fine-tuning performance. We hypothesize that input noise implicitly trains the model to generalize to alternative aliases: for instance, given the mention "Yuri Gagarin", the model might have to learn to recognize "Yuri [MASK]" or "[MASK] Gagarin".

Encouraged by the success of Devlin et al. 2018, we experimented with also pretraining our model with a masked-language modelling objective, with the expectation that such pretraining would help our model learn better representations of language. We tried different architectures and loss weights (including a 4 layer transformer with both objectives at layer 4, a 12 layer with the entity linking loss at layer 4, etc.) but overall found this to not improve further on simply adding noise in pretraining.

| System | No fine-tuning | Fine-tuned |
|---|---|---|
| With noise | 92.2 | 96.9 |
| Without noise | 88.1 | 96.1 |

Table 5: Impact of adding BERT-style input noise during pre-training. We report development accuracy on CoNLL using the Pershina et al. 2015 alias table.

| | None | Title | Title & first two sents |
|---|---|---|---|
| No Fine-tuning | 85.5 | 89.4 | 92.2 |
| Fine-tuned | 96.2 | 96.9 | 96.9 |

Table 6: Impact of additional context beyond a single sentence used for entity disambiguation performance on the CoNLL task. We report development accuracy on CoNLL using the Pershina et al. 2015 alias table.

### 6.4 Impact of context selection methods on CoNLL

Table 6 shows the impact of varying CoNLL's context type in pre-training and fine-tuning. We find that larger contexts, especially those that include the beginning of the document, considerably boost performance before fine-tuning. However, similarly to our observations in Sections 6.2 and 6.3, we find that improvements are less marked after fine-tuning, likely because our performance is already very high.

### 6.5 Error analysis

Figure 2 shows three sample errors on the CoNLL development set. Most errors are due to varying levels of specificity in the CoNLL labels. Some errors are due to changes in Wikipedia. For instance, in text A, the Bulgaria U21 soccer team Wikipedia page was built in 2013, after CoNLL. Also, in text C, our model correctly disambiguates between Austin, Michaella and Richard Krajicek, which are all three tennis players (only Richard is a dutch).

## 7. Conclusion

In this paper we present a thorough study of pretraining strategies for supervised entity linking, achieving state-of-the-art performance on both CoNLL and TAC-KBP 2010 with a four-layer Transformer-based model. Given the limited headroom remaining in these datasets, and the strong impact of alias tables in simplifying the problem, we believe the creation of new datasets, and more difficult entity linking settings, such as zero-shot and low resource domains, are crucial areas for future work.

## Acknowledgments

The authors wish to thank Dan Bikel and Eunsol Choi for their helpful comments in the preparation of this paper, as well as the anonymous reviewers.

---

**Text a**: Soccer - Israel beat Bulgaria in European under-21 qualifier. Herzliya, Israel 1996-08-31. *Incorrectly predicted* "Bulgaria" as `Bulgaria national under-21 football team`. Second prediction is correct: `Bulgaria national football team`.
*Correctly predicted* "Israel" as `Israel national football team`. (There is no Wikipedia page for Israel national under-21 football team page). *Correctly predicted* "European".

**Text b**: Scottish labour party narrowly backs referendum. Stirling, Scotland 1996-08-31. Conservatives have only 10 of the 72 Scottish seats in parliament and consistently run third in opinion polls in Scotland behind labour and the independence-seeking Scottish national party.
*Incorrectly predicted* "labour party" as `Scottish Labour Party`. Gold Label: `Labour party (UK)`.

**Text c**: Edberg refuses to qo (*sic*) quietly. Richard Finn "it doesn't look all that bad" Edberg said of his path through the draw starting next with a match against Krajicek's dutch countryman Paul Haarhuis.
*Correctly predicted* "Edberg" as `Stefan Edberg` and "Krajicek" as `Richard Krajicek` and "Paul Haarhuis".
*Incorrectly predicted* "dutch" as `Dutch people`. Second prediction is `Netherlands` and correct.

---

Figure 2: Sample of errors on the CoNLL development set for our model.

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
