# OpenReview forum: "Empirical Evaluation of Pretraining Strategies for Supervised Entity Linking"
_AKBC.ws/2020/Conference — AKBC 2020_

### Official Review · AnonReviewer2 · 2020-03-27
**thorough analysis of pre-training strategies for transformer-based entity linking**

**Rating:** 7
**Confidence:** 3

**Review:**

The paper describes an evaluation of several pre-training strategies for the task of entity linking, using the AIDA and TAC-KBP baselines. In particular, the authors look at the impact of entity candidate selection strategies, adding noise during pre-training, and context selection methods. The model employed for entity disambiguation is a 4-layer transformer for the language representation, with an MLP final layer to perform disambiguation. The analysis of the pre-training strategies is detailed, and could be interesting for others using the transformer architecture to perform entity linking. Minor issue, but the paper is missing a conclusion section - this could be used to discuss how these results can generalize to other methods for entity linking.

---

> ### Author Response · Authors · 2020-04-08
> **Response**
>
> Thank you for your review. We will incorporate your suggestion about an expanded discussion and conclusion in our final version.

---

### Official Review · AnonReviewer3 · 2020-03-28
**Solidly done piece of empirical work**

**Rating:** 7
**Confidence:** 4

**Review:**

This paper investigates the use of a simple architecture for entity disambiguation: encode the mention and its context with BERT, use an MLP over the mention's fenceposts to compute an embedding, then compare that embedding with embeddings of entity candidates and take the one with the highest dot product. Notably, it uses a transformer pre-trained on Wikipedia to do entity resolution, but does *not* use the BERT model or its pre-trained parameters directly.  The paper deals with several design decisions along the way: how to pre-train this model on Wikipedia, how to generate candidates at train and test time, whether or not to mask the input as in BERT, and other hyperparameters. Results show state-of-the-art performance on CoNLL (with a good candidate set) and TAC-KBP, as well as good performance on end-to-end entity linking (detecting and linking mentions).

This paper isn't exceptionally creative. However, it's a solidly done piece of empirical work that in my opinion should exist in the literature.  While a lot of work has moved onto zero-shot settings (Ling et al./Wu et al./Logeswaran et al. that the authors cite, plus Onoe and Durrett "Fine-Grained Entity Typing for Domain Independent Entity Linking") or other embedding-based formulations (Mingda Chen et al. "EntEval: A Holistic Evaluation Benchmark for Entity Representations"), a strong, conventional, up-to-date supervised baseline should exist in the literature and currently doesn't.

The one idea here that seems unconventional is foregoing BERT-based pre-training and only pre-training on the entity linking task itself.  This is an interesting choice but I'm not too surprised it works well: Wikipedia is already pretty big, and this approach lets you learn good entity embeddings in the same space as the transformer encoder.

The experiments in this paper are quite well-done and touch on a lot of issues surrounding how the system is trained.  I'm glad to see the authors use the TAC-KBP data and start to make this more standard -- it would've been nice to see other datasets like WikilinksNED or some of the older/smaller datasets from Ratinov et al. (2011) "Local and Global Algorithms for Disambiguation to Wikipedia"). The CoNLL data is weird and limited in scope.  Nevertheless, achieving state-of-the-art on this well-worn dataset is impressive.

Table 4 was probably the most surprising part of the paper to me.  It's a little strange that the OURS candidate selection method works poorly on TAC-KBP. It basically seems like a union of phrase table and page, right?  I understand the paper's high-level point that random is somehow closer to the true TAC-KBP task, but this argument seems handwavy and doesn't seem like it should make such a large difference.

BERT-style noise is also surprisingly effective during pre-training. The paper's interpretation of this makes sense.

Overall, I feel like this paper deserves to be published: the results will be a good benchmark for future efforts and I can imagine other researchers using this as a starting point.

---

> ### Author Response · Authors · 2020-04-08
> **Response**
>
> Thank you for your comments. In regards to Table 4, since no alias table is used for inference in the TAC-KBP dataset, we believe a random sample of negatives is closer to the full softmax used at inference time, whereas the biased sample introduced by candidates is better suited to situations where an alias table is present. However, we agree that further investigation would be required to confirm this intuition.

---

### Official Review · AnonReviewer1 · 2020-03-29
**useful analysis of pretraining strategies for supervised entity linking**

**Rating:** 7
**Confidence:** 4

**Review:**

This paper presents an empirical study of pretraining strategies for supervised entity linking. Previous works either focus on constructing general-purpose entity representations or zero-shot entity linking and do not fully explore pretraining. The  paper is well written and is easy to follow. I think the findings in the paper should be of interest to the AKBC community.

The proposed model achieves competitive performance even without domain-specific tuning. A detailed empirical analysis of negative candidate selection, noise addition, and context selection is presented.  The proposed model is able to perform  end-to-end entity linking with simple modeling and low inference cost.

Missing comparison with related work on end-to-end entity linking with BERT:
Investigating Entity Knowledge in BERT with Simple Neural End-To-End Entity Linking, Samuel Broscheit, CoNLL'19

A figure / running example to illustrate where the demonstrated benefits of pretraining come from over prior art will strengthen the paper.

A discussion on potential limitations of pretraining will also be informative.

---

> ### Author Response · Authors · 2020-04-08
> **Response**
>
> Thank you for your comments and for pointing us to this related work. We will add an explicit comparison to Broscheit’s work and incorporate your suggestions for improved clarity in our final version.

---

### Public Comment · ~Samuel_Broscheit1 · 2020-04-02
**Comparison to "Investigating Entity Knowledge in BERT with Simple Neural End-To-End Entity Linking"**

Dear authors,

"Investigating Entity Knowledge in BERT with Simple Neural End-To-End Entity Linking" from CONLL 2019 https://arxiv.org/pdf/2003.05473.pdf investigated finetuning/pretraining on Wikipedia for End-To-End Entity Linking and achieved better performance (79.3 vs 76.7 Micro P@1) on AIDA-CONLL. Not using BERT's pretrained parameters or the lower number of parameters in your model could explain the performance difference.

It would be interesting if you could discuss the differences to this previous work. A contribution of your study certainly is to show how much the use of candidate reduction through the alias table and adding of input noise helps. As far as I understood you do domain-specific tuning on AIDA-CONLL, correct? (this is not clearly explained, or I missed it)

Thank you

---

> ### Author Response · Authors · 2020-04-08
> **Response**
>
> Thank you for pointing us to this work, we will be sure to include a comparison and discussion in our final paper. We do finetune our model on each of the final datasets -- this will be made clearer.

---

### Decision · Program_Chairs · 2020-04-30

**Decision:**

Accept

**Comment:**

All reviewers agreed that the paper has some strengths with merits outweighing (a few) flaws.

This paper investigates the use of a simple architecture for entity disambiguation, while exploring several design decisions along the way. Results show state-of-the-art performance on CoNLL (with a good candidate set) and TAC-KBP, as well as good performance on end-to-end entity linking (detecting and linking mentions).

The strengths of this paper are: (1) competitive performance without domain-specific tuning, (2) extremely well done experiments touching on many related issues (negative candidate selection, noise addition, and context selection). One of the reviewers describes it as "solidly done piece of experimental work", which "will be a good benchmark for future efforts".

There are two drawback of the paper. (1) the techniques in this paper by themselves aren't novel. In fact, one cannot attribute a strong technical contribution for this paper. So, if one has to accept the paper it has to be for experiments and analysis and not for the novelty. (2) there is another paper from CONLL'19 which is related. The reviewers liked the experiments in this paper better than the CONLL paper, which are much more thorough in a wider range of experimental settings.